# Non-invasive focusing and imaging in scattering media with a fluorescence-based transmission matrix

Antoine Boniface [1 ✉], Jonathan Dong[1,2] & Sylvain Gigan [1]

In biological microscopy, light scattering represents the main limitation to image at depth. Recently, a set of wavefront shaping techniques has been developed in order to manipulate coherent light in strongly disordered materials. The Transmission Matrix approach has shown its capability to inverse the effect of scattering and efficiently focus light. In practice, the matrix is usually measured using an invasive detector or low-resolution acoustic guide stars. Here, we introduce a non-invasive and all-optical strategy based on linear fluorescence to reconstruct the transmission matrices, to and from a fluorescent object placed inside a scattering medium. It consists in demixing the incoherent patterns emitted by the object using low-rank factorizations and phase retrieval algorithms. We experimentally demonstrate the efficiency of this method through robust and selective focusing. Additionally, from the same measurements, it is possible to exploit memory effect correlations to image and reconstruct extended objects. This approach opens up a new route towards imaging in scattering media with linear or non-linear contrast mechanisms.

[1] Laboratoire Kastler Brossel, Sorbonne Université, École Normale Supérieure-Paris Sciences et Lettres (PSL) Research University, Centre National de la Recherche Scientifique (CNRS) UMR 8552, Collège de France, 24 rue Lhomond, 75005 Paris, France. [2] Laboratoire de Physique de l'École Normale Supérieure, Université Paris Sciences et Lettres (PSL), Centre National de la Recherche Scientifique (CNRS), Sorbonne Université, Université Paris-Diderot, Sorbonne Paris Cité, 24 rue Lhomond, 75005 Paris, France. ✉email: antoine.boniface@lkb.ens.fr

Propagation of light in materials with refractive index inhomogeneities, such as biological tissues, results in scattering. In such disordered media, ballistic light exponentially decreases with penetration depth, which limits the scope of conventional optical microscopy. At depth, coherent light is affected by multiple scattering and produces very complicated interference patterns, known as speckle[1]. Recent years have witnessed many advances in the ability to coherently manipulate this figure of interferences owing to the availability of spatial light modulators (SLMs)[2,3]. In particular, they enable refocusing light to a diffraction-limited spot[4]. These techniques often rely on optimizing the incident wavefront such that it maximizes a feedback signal emitted from the target focus point. A key constraint for biological imaging at depth, is that the measurement of the feedback has to be non-invasive. Several strategies using acoustics or non-linear fluorescent guidestars have been proposed for this purpose[5]. However, most of them are only able to form a single focus at a given output position[6–9], which limits the acquisition speed or the field-of-view.

Deterministic focusing of light on multiple targets is optimally achieved with a transmission matrix (TM) that linearly relates the input field to the output field[10]. However, its non-invasive measurement remains very challenging. Indeed, even if each target has its own optical response, what is measured in epi-detection is the back-scattered emission, thus spatially and temporally mixed. To overcome this limitation, optics has been combined with acoustics to coarsely locate each target[11,12], which enables the reconstruction of a TM but requires complicated acousto-optical setups. Another powerful approach relies on the measurement of a time-gated matrix in reflection[13,14], but is based on retro-reflected ballistic photons, hence limited in depth.

Linear fluorescence remains an essential technique in microscopy because systems are fairly inexpensive and easy to handle. As such, it remains a staple tool in biology and biomedical sciences. It has enabled imaging of cells and sub-microscopic cellular components with high spatial resolution, specificity, contrast and speed. Combined with light-sheet or structured illumination microscopy, linear fluorescence allows sectioning and imaging at moderate depth[15,16]. Although imaging fluorescent objects through thin scattering media can be done thanks to the memory effect[17,18], a general method to focus on fluorescent objects at depth, and image them if they extend beyond the memory effect is still missing.

Here, we report on a robust TM approach for fluorescence imaging through a relatively strong scattering medium, in a non-invasive way. The technique relies on shining a sequence of known wavefronts on a fluorescent object hidden behind a scattering medium, and collect in reflection the corresponding low-coherence fluorescent speckles back-scattered by the medium. From this set of input-output information, we are able to computationally retrieve both (i) the ingoing field-TM for the excitation light, and (ii) the outgoing intensity-TM for the fluorescence light. We demonstrate robust and selective focusing across all the object, both on beads and on more complex fluorescent objects. Finally, if the medium exhibits limited memory effect, we show that an image of the object can be retrieved, even when its size exceeds the memory effect range.

## Results

The experimental apparatus is depicted in Fig. 1a. A coherent beam of light is first modulated in phase by an SLM and directed through a scattering medium onto a fluorescent object made of several emitters. To describe both the ingoing and outgoing light propagation, we use a transmission matrix formalism. A field-TM, denoted $T$, connects the input field $E_{in}$ (specifically the phase pattern displayed onto the SLM) to the field at the position of the N targets. Thus, the speckle intensity in the plane of the fluorescent object reads $|E_{exc}|^2 = |TE_{in}|^2$. Once excited, each target fluoresces proportionally to its illumination. This low-coherence signal is back-scattered by the medium and can be non-invasively measured with a camera placed in reflection. It can be written as $I_{out} = W|E_{exc}|^2$, where $W$ is an intensity-TM, linking the N targets to the $D$ pixels of the camera via their respective fluorescent eigen-patterns. We define here eigen-patterns, as all the independent speckles, each single target generates on the camera. It is worth stressing that the measurement is made in reflection only, thus entirely non-invasive. A control camera placed on the far side of the sample allows to monitor the excitation patterns $|E_{exc}|^2$ at the object plane.

Our technique relies on exciting the sample consisting of the scattering medium and the fluorescent object with a variety of $p = 1, …, P$ random input phase patterns $E_{in}(p)$ and collecting the fluorescence responses $I_{out}(p)$ reflected on the same side. For all $p = 1, …, P$, $I_{out}(p)$ can be written as:

$$I_{out}(p) = W|E_{exc}(p)|^2 = W|TE_{in}(p)|^2 \tag{1}$$

$I_{out}(p)$ corresponds to a low contrast speckle because, first, the fluorescence emission is broadband and not polarized, second, the N beads generate N different speckles that partially average out[1]. Nevertheless, the decrease in contrast due to the number of beads is relatively slow and scales as $\sqrt{2/N}$ in the case of linear fluorescence (see Supplementary Note 8). The overall output $I_{out} \in \mathbb{R}_+^{D \times P}$ can be written as a rank-N product of two positive matrices $I_{out} = WH$ (with $N \ll D$, $P$), where both $H = |TE_{in}|^2$ and $W$ are real positive matrices. This corresponds exactly to the framework of Non-negative Matrix Factorization (NMF) that we, therefore, use to estimate $W$ and $H$ from $I_{out}$, see Fig. 1b. Thanks to its robustness and interpretability, this framework has already been applied in many settings[19], including the fluorescent readout of neuronal activity[20,21], but has never been associated with wavefront shaping yet. It is interesting to note that similar approaches based on leading eigenvector decompositions (without the non-negativity assumption) have already been used in optics[22] and in combination with acoustics[23] for imaging at depth. In a second step, a phase retrieval (PR) algorithm allows retrieving the ingoing field-TM, $T$[24,25], from $H$. This computational problem has been studied extensively for the case of random matrices and is in theory solvable when the number of measurements (here, $P$) is a few times larger than the dimension of the unknown (here, $N_{SLM}$)[26–28].

Experimentally, we first performed the measurement with fluorescent objects made of 1μm beads, placed behind holographic diffusers. Once a series of input patterns have been displayed and the corresponding fluorescence images recorded, the matrix $I_{out}$ is factorized into two low-rank matrices thanks to NMF. The rank $r$ of the factor matrices is the main input parameter required to run the algorithm. In principle, $r$ should correspond to the number of independent sources $N$ in the system, which in this specific case corresponds to the number of resolvable targets $N$ since the speckle grain matches the size of the bead. Note that $N$ is unknown in such reflection configuration. Nevertheless as discussed in Methods, an upper bound for $r$ can be easily estimated from $I_{out}$, and is sufficient to identify all the N targets. An additional step of phase retrieval estimates the field-TM $T$ that links the SLM pattern to the plane of the beads.

Focusing light using phase conjugation provides a method to check the quality of the NMF + PR pipeline. On Fig. 2, this focusing capability is shown with images monitored on the control camera, but also non-invasively from the observation of back-scattered fluorescence. When light is successfully focused on

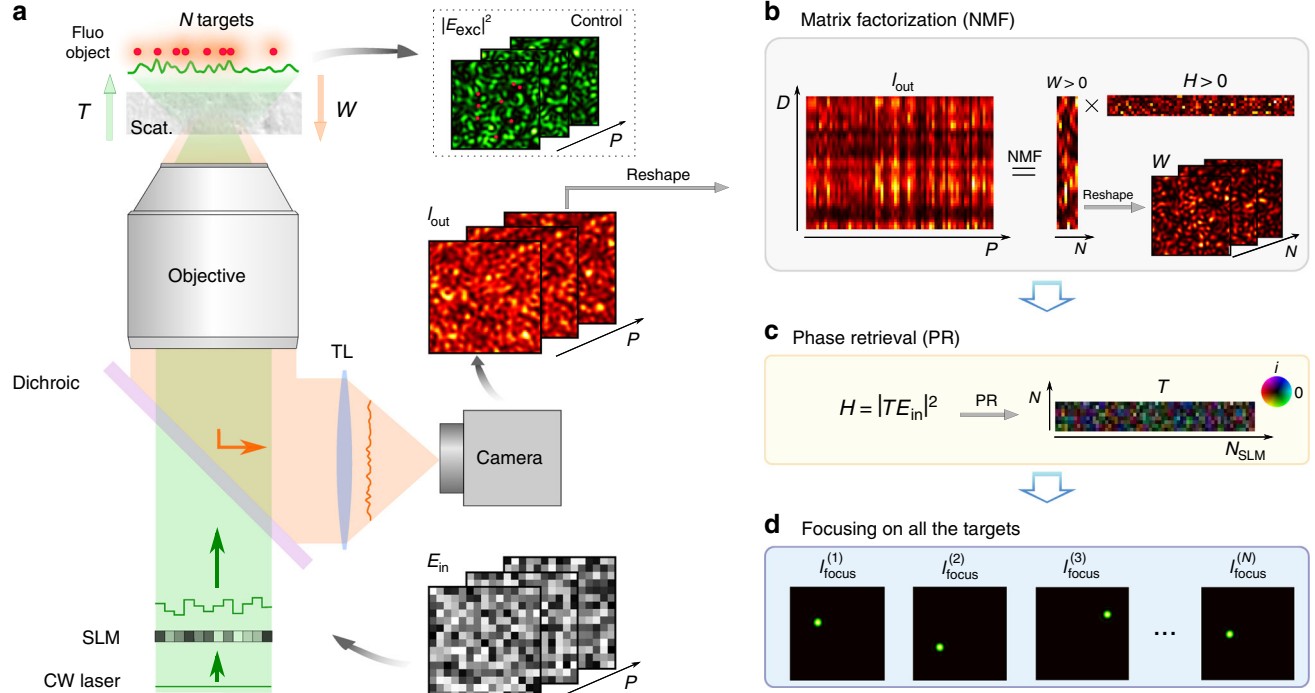

**Fig. 1 Double-TM reconstruction principle - simulation results. a** Schematic view of the experimental setup. Coherent light is sent on a fluorescent object, made of $N$ targets, hidden behind a scattering medium. A given speckle field, $E_{exc}(p) = TE_{in}(p)$, illuminates the object which emits a fluorescence signal in return, with $p = 1, ..., P$ the index of the SLM pattern. A portion of the latter is back-scattered by the medium and epi-detected on a camera, $I_{out}(p) = W|E_{exc}(p)|^2$. TL tube lens, Scat. scattering medium. **b** $I_{out}$ is a sequence of $P$ fluorescent speckles recorded for different inputs, $E_{in}$. This matrix admits a rank-$N$ factorization $I_{out} = WH$, where $W$ and $H$ are unknown positive matrices. NMF is used to retrieve them. $W$ is an intensity-TM describing the fluorescence propagation from the object to the camera. **c** $H$ describes light propagation from the SLM to the object and can be written as $H = |TE_{in}|^2$, where $T$ is a field-TM. An additional step of phase retrieval gives access to $T$. **d** Phase conjugation of $T$ is used to selectively and non-invasively focus light on all the targets of the fluorescent object. This focusing ability is used to quantify the quality of the double-TM reconstruction.

a target, the spatial variance of the fluorescence speckle increases[7]. Since we typically overestimate the rank $r > N$, the NMF may generate spurious eigen-patterns (which do not focus the illumination) and duplicates. Looking at the spatial variance and the correlation between fluorescent patterns allows to identify both (see Supplementary Note 4). We thus validate the ability to accurately reconstruct the ingoing and outgoing transmission matrices and deterministically focus on every beads.

The double-TM reconstruction can be applied in principle whatever the depth and scattering properties of the medium, as long as it provides a measurable fluorescent speckle. In the following, we show that, if there is some memory effect (ME), the technique allows not only focusing but also fluorescence imaging at depth by looking at the correlations between fluorescent eigen-patterns.

In essence, two beads within the ME should exhibit translated fluorescent patterns with a shift equal to their relative distance. By cross-correlating the fluorescence patterns, which are recorded in epi while displaying the $r$ focusing patterns onto the SLM, it is possible to retrieve a distance map between all the beads. In Fig. 3, we show an example of such reconstruction with an object much larger than the ME range (see Supplementary Note 5). Interestingly, computing successively these pairwise correlations between close targets allows retrieving the full object, well beyond a single ME patch. All the beads are thus reconstructed as long as their ME patches have some overlap. In Fig. 3 we show reconstruction of an object extending approximately three times the ME range. Note that using directly the $W$ patterns from the NMF works but did not provide here as good results (see Supplementary Note 6).

Finally, to demonstrate that our technique can also be used with continuous volumetric objects, we tested it on biological objects, here fluorescence-stained pollen grains. The whole process, from the acquisition to the reconstruction, is similar to what is presented in Fig. 3. Figure 4a–d shows fluorescent images of three different pollen grains taken without the diffuser. The blue lines are contours of the reconstructed image at 10% of the maximum intensity, showing that the high-intensity features of the object are faithfully retrieved. The full reconstructed images are presented in Fig. 4e–h. While the reconstruction appears grainy, we do retrieve the main and brightest features of the pollen seeds. With $N_{SLM} = 256$ pixels (Fig. 4e–g), the SNR of the focus is not very high, around 10–20, but sufficient to connect the epi-detected fluorescent speckle to the eigen-pattern of the focused emitter and reconstruct the shape of the pollen seeds. Increasing the number of SLM pixels to $N_{SLM} = 1024$ as in Fig. 4h did not improve significantly the results.

## Discussion

Several conditions should be met in order to successfully operate our non-invasive technique. A first point is the required number of patterns to accurately retrieve the double-TM. As detailed in Supplementary Note 3, reconstructing $W$ can be done even with a low number of patterns related to the complexity of the object (number of separate emitters). On the other hand, $T$ is recovered through an additional step of phase retrieval which requires a larger number of patterns, related to the number of SLM pixels. For example, experiments of Figs. 2 and 3 require $P \simeq 15{,}000$ patterns, which at 50 Hz (limited by the exposure time of the camera) corresponds to ~5 min. Diminishing the number of SLM pixels as in Fig. 4 where $N_{SLM} = 256$ reduces

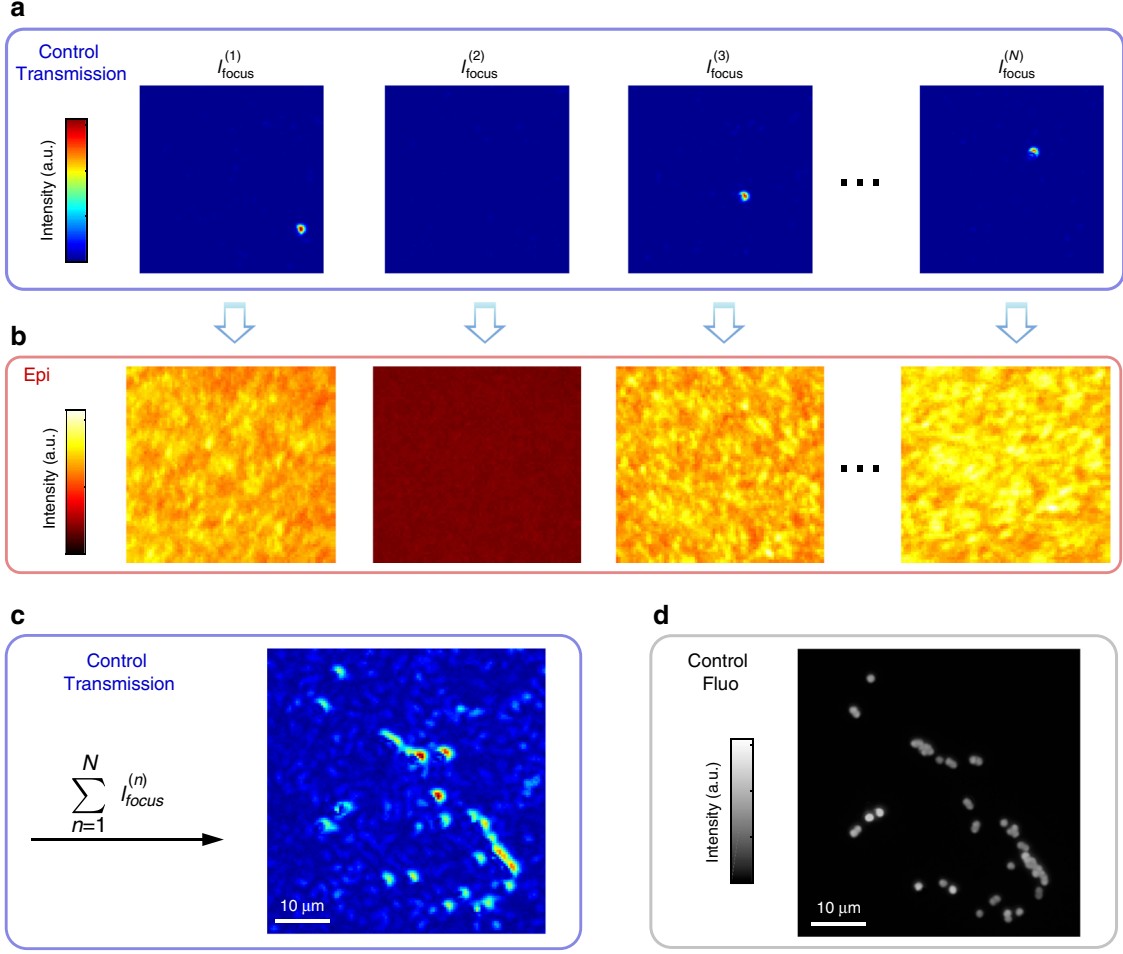

**Fig. 2 Experimental Results: TM reconstruction and focusing on beads through a ground glass diffuser.** From the experimentally measured $I_{out}$, the field-TM $T$ is reconstructed using NMF and a phase retrieval algorithm. To prove the success of the reconstruction, we investigate in the ability of $T$ to focus excitation light. A rank $r = 30$ is set as input parameter for the NMF. **a** Examples of focus spots obtained after phase conjugation of $T$. Since the rank is overestimated ($r > N$), the NMF may generate spurious eigen-patterns and reconstruction thus partially fails. Corresponding points do not generate a focus, like $I_{focus}^{(2)}$. **b** This validation step can be performed non-invasively by looking at the spatial variance of the epi-detected fluorescent patterns. **c** Sum of all the $r = 30$ intensity images recorded on the control camera. It shows that our technique is able to focus on most targets. **d** Fluorescence image of the object obtained without the scattering medium for control only. Here $P = 15,360 - $ Acq. speed $= 50\,Hz - N_{SLM} = 1024$.

the acquisition time to few tens of seconds, which should be compatible with stability time of ex-vivo biological tissues[29].

Another important aspect is the complexity of the object that can be reconstructed. Here we demonstrate focusing and imaging on multiple beads, but also on continuous and even volumetric objects. Reconstructions up to around 50 focus positions have been performed in this work. One limitation is the contrast of the measured speckle, that decreases with the complexity (number of separate emitters) of the object, but only with a mild squareroot dependency. For further insight, we may refer to the Supplementary Information of[7] where a study about the contrast measurement in presence of noise was carried out and[19] where the effect of noise on the NMF is investigated. In tissues, a general problem is the background fluorescence, that could be tackled via appropriate sparse staining or acoustic tagging of a small region[30,31].

Regarding imaging, our technique is limited by the spatial sparsity of the object rather than its size that can be much larger than the memory effect range. We propose in the Supplementary Note 7, an alternative algorithm based on Multi-Dimensional Scaling (MDS) that offers some advantage in terms of noise robustness. But still, the major limitation is that isolated targets (without correlations with others) cannot be correctly located.

It is important to note we only reconstruct a 2D projection of an object, even though the object might be volumetric. Strategies for 3D reconstruction after propagation in scattering media have been proposed[32,33], but require specific experimental configurations. Additionally, techniques to reduce background fluorescence[30,31] may be useful to reduce the necessity of a full 3D reconstruction.

We focused here on linear fluorescence contrast, but the technique should readily generalize to any incoherent linear mechanisms, such as spontaneous Raman. Non-linear incoherent processes should also be possible (as shown in Supplementary Note 8 for 2-photon fluorescence), which should benefit from a higher contrast and lower background, at the cost of a lower overall signal.

In conclusion, we have presented a completely non-invasive computational strategy to characterize light propagation in and out of a scattering medium based on linear fluorescence feedback only. It allows both focusing at depth and, providing some memory effect is present, imaging of an extended object. The

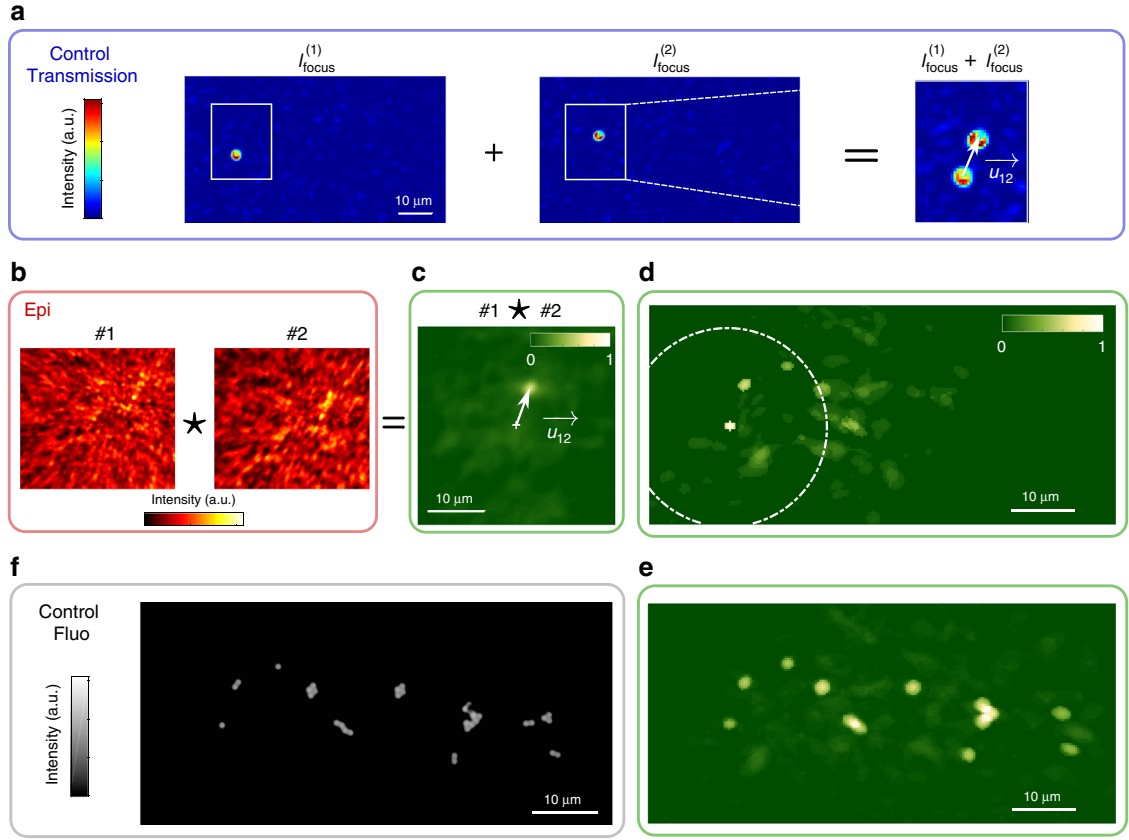

**Fig. 3 Imaging through scattering media using back-scattered fluorescence and finite memory effect. a** Images recorded on the control camera when the illumination is focused on two different beads of an extended object represented in **f**, where the scattering medium is made by two surface diffusers separated by 0.78 mm. We define $\vec{u_{12}}$ as the relative displacement between the two foci (i.e. beads). **b** If the two beads are within the same ME patch their two fluorescent patterns, #1 and #2, are spatially shifted by $\vec{u_{12}}$. **c** $\vec{u_{12}}$ is estimated from cross-correlation #1⋆#2. **d** This operation is repeated between #1 all the other eigen-patterns #$j$ with $j=1, …, 13$. If bead #$j$ is within the ME patch of #1, indicated by the dashed circle, a peak of correlation appears and a sub-part of the object is retrieved. **e** To obtain the full extended object, the reconstruction is done as in **d** for all the ME patches. The result is in good agreement with the ground truth (**f**) which is a fluorescence image recorded without scattering medium. $P = 14{,}336 -$ Acq. speed $= 20$ Hz $- N_{SLM} = 1024$.

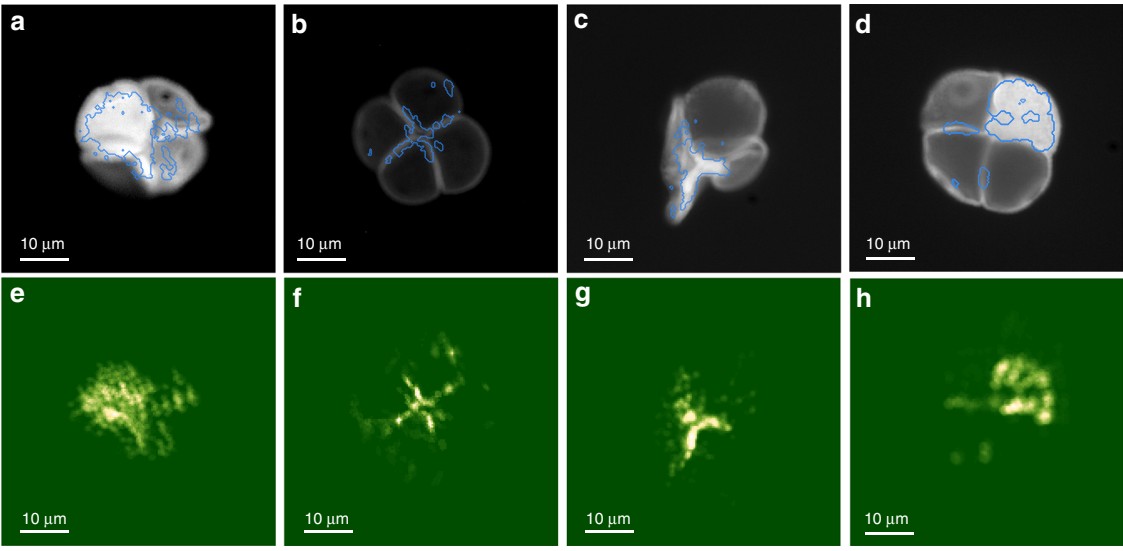

**Fig. 4 Reconstruction examples for continuous volumetric fluorescent objects through a ground glass diffuser. a–d** Fluorescence images of different pollen seed structures taken after removing the diffuser. The blue line is a contour plot (at 10% of the maximum intensity) of the reconstructed object. **e–h** Reconstruction of the object with NMF+PR approach. The rank for the NMF is overestimated at $r=80$ for **e–h** and $r=150$ for **f–g**. Note that only the high variance eigen-patterns, are cross-correlated. In the four cases, $P = 5120$ fluorescent images are recorded at 50 Hz, $N_{SLM} = 256$ for **e–g** and $N_{SLM} = 1024$ for **h**.

method is very simple, robust, and provides a promising route towards deep fluorescence imaging beyond the ballistic regime. It should be applicable to a large variety of contrast mechanisms.

## Methods

**Experimental setup**. A continuous-wave laser ($\lambda = 532$ nm, Coherent Sapphire) is expanded on a phase-only MEMS SLM (Kilo-DM segmented, Boston Micromachines), such that all the $N_{SLM} = 1024$ segments can be used. Once modulated, the beam is directed through the illumination objective (Zeiss W "Plan-Apochromat" 20× , NA 1.0) to excite the fluorescent object made of orange beads (540/560 nm, Invitrogen FluoSpheres, size 1.0 µm) or pollen seeds (Carolina, Mixed Pollen Grains Slide, w.m.) placed on top of the scattering medium. The excitation beam (diameter < 6 mm) underfills the objective back aperture (diameter 20 mm) which reduces the actual illumination NA. It results that the speckle grain size at the fluorescent object plane is around 1 µm. The SLM is imaged to the back focal plane of the microscope objective. The scattering medium is not the same in all the experiments in order to control the memory effect. In the experiment presented in Fig. 2 we use a ground glass (Thorlabs, DG10), in Fig. 3 we use two holographic diffusers (Newport 1° + Newport 10°) and in Fig. 4 only one holographic diffuser (Newport 1°).

Part of the 1-photon fluorescence emission is back-scattered by the medium and epi-detected on a first camera: CAM1 (sCMOS, Hammamatsu ORCA Flash). Recording of the $P$ fluorescence images is the slowest step throughout the acquisition process; it is between 20 and 50 Hz depending on the scattering medium and the fluorescent sample. Once acquired, raw images are cropped (such that one image contains roughly few tens of speckle grains). Then a high pass Gaussian filter removes the background which significantly improves the contrast. The corresponding data form a matrix $I_{out}$ which is later processed with the algorithm to reconstruct the two TMs. We use a dichroic mirror shortpass 550 nm (Thorlabs) and two other filters (F): a 532 nm longpass (Semrock) and a 533nm notch (Thorlabs). Additional spectral filters can be used before light reaches CAM1 in order to narrow the spectral width of the detected fluorescence and increase the contrast of the fluorescent speckle. To the same end, a polarizer can be inserted, enhancing the contrast by a factor $\sqrt{2}$. A second microscope objective (Olympus "MPlan N" 20×, NA 0.4), placed in transmission, provides an image of the plane of the beads, onto a CCD camera CAM2 (Allied Vision, Manta). This part of the setup is for passive control only. It allows us to correctly position the beads using a white light source (Moritex, MHAB 150 W), but also to monitor illumination speckles $|E_{exc}|^2$.

In the first experiment presented in Fig. 2, $P = 15,360$ different random inputs are generated and corresponding fluorescence images of size $D = 50 \times 52 = 2600$ pixels were recorded. In the second experiment presented in Fig. 3, $P = 14336$ different random inputs were generated and corresponding fluorescence images of size $D = 70 \times 64 = 4480$ pixels were recorded on the camera in epi. In the third experiment presented in Fig. 4, $P = 5120$ different random inputs were generated and corresponding fluorescence images were recorded on the camera in epi.

The experimental setup is shown in Supplementary Note 1.

**NMF + PR algorithm**. Before factorizing $I_{out}$ with the NMF algorithm, the rank $r$ of the low-rank factor matrices needs to be determined. It is related to the number of fluorescent beads in the sample and is not known in our reflection configuration. We estimate it by looking at the residual error $||I^{fluo} - WH||_F$ as a function of the rank $r$. Its plot should have a typical change of slope, as described in ref. [34]. It provides a good estimate for the rank of $I_{out}$. However, when the number of targets $N \gtrsim 10$ we experimentally observe that the change of slope cannot be determined with good accuracy. As detailed in Supplementary Note 2, we decided to take the upper bound and remove spurious values afterwards.

For the NMF, we use the *nnmf* Matlab function with default parameters. In particular, matrices for the initialization are random.

For the PR we use an algorithm very similar to ref. [26], involving a spectral method to obtain a good initial estimate for the subsequent gradient descent iterations, except that we use a refined spectral initialization to speed up the convergence[27,28].

Simulation codes are available at: https://github.com/laboGigan/NMF_PR

## Data availability

All relevant data are available from the authors upon request.

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

## Acknowledgements

We thank Claudio Moretti for fruitful discussions and constructive comments. This research has been funded by the European Research Council ERC Consolidator Grant (Grant SMARTIES - 724473). S.G. is a member of the Institut Universitaire de France.

## Author contributions

A.B., J.D. and S.G. conceived the idea. A.B. and J.D. wrote the Matlab code to collect and process experimental data. A.B. performed the experiments and analysed the experimental data. All authors discussed the results and commented on the paper.

## Competing interests

The authors declare no competing interests.
