## [Peer Review File · Nature Communications]

Reviewer #1 (Remarks to the Author):

The manuscript "Non-invasive focusing and imaging in scattering media with a fluorescence-based transmission matrix" by Boniface et al. reports a non-invasive method to image fluorescence targets through scattering media by solving the transmission matrix of the scattering media. The authors conducted an experimental demonstration to image the fluorescence samples through thin and static samples. In brief, the method randomly realizes the input fields to excite the fluorescent objects through a piece of scattering medium and measures a set of fluorescence emission patterns in back reflection geometry. By implementing non-negative matrix factorization to the matrix constructed by the set of the intensity patterns, the authors were able to retrieve the "eigen-patterns" from individual fluorescence targets as well as the underlying patterns on the fluorescence targets. A phase retrieval algorithm was then implemented to retrieve the complex transmission matrix of the scattering medium based on the retrieved intensity patterns on the fluorescence targets. Using this transmission matrix, the authors focused light to the fluorescence targets sequentially and separately. To form an image of the fluorescence targets, the "memory effect" based approach is exploited at the subsets of the correlated emitted patterns. A large field of view of the image is then obtained by stitching the reconstructed images patch by patch.

The method is non-trivial and elegant, and the experimental results look great and convincing. There are several shortcomings of this method in terms of broad applications in living samples. First, "memory effect" is required to form an image, which limits the imaging depth; second, the sample needs to be static at least for tens of seconds; third, sparsity is preferred for a good reconstruction. These are the hard problems in fluorescence imaging deep inside living samples. Although there is still a long way for this technique to be practically useful, the method is clearly an important addition to the toolbox for tackling the problem of optical scattering, and the work brings in new and intriguing thoughts to the research field. I think the work will be of great interest to the readers of Nature Communications. I am happy to recommend the publication of the work.

Here are some comments that the authors may want to consider addressing before publication.

- On page 3, "In principle, r should correspond to the number of independent targets N in the system, which is unknown in such reflection configuration." Does the speckle size on the target play a role here? Does the rank of the matrix depend on the number of the speckles on the fluorescence targets? If so, it may be more accurate to describe the target population in terms of the number of speckles. If there is only one target that covers multiple speckles, what is the rank of the matrix? Can the method achieve a single speckle focusing resolution in this case?
- It would be helpful if the authors can provide an estimation of the population limit of fluorescence targets under which the image can be successfully reconstructed.
- In the caption of Fig. 1. "A speckle field, $E_{exc} = TE_{in}$, illuminates the object which emits a fluorescence signal in return." The dimension of E_{in} is not explicitly described. This sentence implies that E_{in} is a vector, i.e. one input field. However, in a couple of other places in the manuscript such as in Fig. 1, " $H = |TE_{in}|^2$ ", E_{in} looks like a matrix consisting of a set of input fields because H is a matrix. It would be great if the authors can clarify the notation.
- On page 4, "FIG. 4. Reconstruction examples for continuous and 3D fluorescent objects through a ground glass diffuser." What is the difference between using 3D and 2D samples? How 3D is defined here? Are the fluorescent objects extended larger than the speckle grain size in the axial direction? If not, the samples may be more precise to be classified as 2D objects in the context of this method.
- On page 1, "Although imaging fluorescent objects through thin scattering media can be done thanks to the memory effect [17, 18], a general method to focus and image a fluorescent object at depth is still missing." This sounds like the method here does not require memory effect, but I think the demonstrated method also relies on memory effect for imaging.
- It seems the retrieved ingoing and outgoing transmission matrices are independent of each other

in the reported method. In terms of hardware implementation, the camera and SLM do not need to be aligned pixel to pixel. However, if the camera and SLM are precisely aligned like the digital optical phase conjugation system, is there any benefit we can gain from this implementation? For example, additional constraints on the transmission matrices for a better reconstruction.

- This method of using NMF to retrieve eigen-patterns shares some similarities with the algorithms used in TROVE (Nature Photonics volume 7, pages300–305 (2013)) and AOTM (ref 12), which are also used to retrieve "eigen-patterns". Could the authors elaborate on the common and different points of these algorithms?

Reviewer #2 (Remarks to the Author):

The authors describe a method to image fluorescent objects hidden behind a scattering medium by using only epi detection

This is a topic of high importance for optical imaging in tissue. A previous attempt by Mosk and colleagues for imaging using only epi detection was limited to the memory effect of the medium, which can be very small.

Here the authors propose a method which goes beyond the memory effect range. This is quite an advance in the field.

It is in many ways quite different from the said previous attempt. The detection uses a 2D camera instead of a single pixel detector and more importantly makes use of the fluorescent speckle pattern.

The method is novel and merit publication.

The manuscript is very well written and thorough.

The fluorescent beads are all placed in the same plane. What happened if they were placed at different depth ? This is a situation that would occur in practice for imaging an unknown sample. The authors hint that the method could be used in 3D.

Wouldn't such a linear technique be limited to the surface of the fluorescing tissue because of the double pass (loss of incident intensity reaching deeper layers and loss of fluorescent intensity on the way back)?

It seems that the power of the technique is that it focuses all the incident light on a single bead. Would confocal detection using ballistic return photons be possible ?

Reviewer #3 (Remarks to the Author):

The manuscript by Boniface and coworkers follows in the steps of a previous manuscript by the same group (Ref. 7) which showed the utility of characterization and maximization of the contrast of the low-coherence fluorescence speckle as a means to focus light in scattering media. Now the authors replace the adaptive search for focusing on individual emitters by a non-negative matrix factorization approach which enables retrieval of the transmission matrix of both the ingoing coherent light and the outgoing fluorescence (at least for the center wavelength for the latter). The beauty of this approach is that it enables to perform the measurements in a wide field, reducing dramatically the time, and cleverly utilizing the multiple fluorescent emitters as witnesses in the measurement of the excitation field transmission matrix. As this approach is new and has exciting potential consequences, it can warrant publication in Nature Communications provided that the authors address the following points:

1. The explanation in the introduction is written in a confusing way. The measured matrix H is low rank because the fluorescent "witnesses" are sparse. reconstruction of the transmission matrix thus seems like magic. It is only much later in the text that the authors actually cite the number of

SLM patterns required to perform the retrieval of the transmission matrix. While this appears in Fig. 1c, it should be much more explicitly stated already when describing the method since it is probably the most significant issue.

2. Relating to the same point, could the authors compare the retrieved images as a function of the number of degrees of freedom in the SLM used in getting the transmission matrix? This would correspond to the tradeoff between measurement time and output quality which is an important parameter.

3. The authors discuss in detail the dependence on the emitter number (and the square root decline in speckle contrast). They do not discuss the effect of the bandwidth of emitted fluorescence. If anyone is to use this type of method, it should be clearly stated (preferably quantitatively) how much the performance deteriorates for dyes or stains with broader fluorescent emission.

4. As can be seen from Fig. 4 (and since the authors assume low rank matrices) the method inherently introduces graininess in the retrieved images even when it is not actually present in the samples. The authors should at least comment on this. Better is if they have a solution to differentiate between an intrinsically grainy sample (like the beads) and a smooth one.

5. All the objects used here are thin objects behind non-fluorescent diffusers. Thus, they do not represent a full 3D problem. Are the claims that this can really work in 3D samples really substantiated?

Author's Response To Reviewer 1

The manuscript "Non-invasive focusing and imaging in scattering media with a fluorescence-based transmission matrix" by Boniface et al. reports a non-invasive method to image fluorescence targets through scattering media by solving the transmission matrix of the scattering media. The authors conducted an experimental demonstration to image the fluorescence samples through thin and static samples. In brief, the method randomly realizes the input fields to excite the fluorescent objects through a piece of scattering medium and measures a set of fluorescence emission patterns in back reflection geometry. By implementing non-negative matrix factorization to the matrix constructed by the set of the intensity patterns, the authors were able to retrieve the "eigen-patterns" from individual fluorescence targets as well as the underlying patterns on the fluorescence targets. A phase retrieval algorithm was then implemented to retrieve the complex transmission matrix of the scattering medium based on the retrieved intensity patterns on the fluorescence targets. Using this transmission matrix, the authors focused light to the fluorescence targets sequentially and separately. To form an image of the fluorescence targets, the "memory effect" based approach is exploited at the subsets of the correlated emitted patterns. A large field of view of the image is then obtained by stitching the reconstructed images patch by patch.

The method is non-trivial and elegant, and the experimental results look great and convincing. There are several shortcomings of this method in terms of broad applications in living samples. First, "memory effect" is required to form an image, which limits the imaging depth; second, the sample needs to be static at least for tens of seconds; third, sparsity is preferred for a good reconstruction. These are the hard problems in fluorescence imaging deep inside living samples. Although there is still a long way for this technique to be practically useful, the method is clearly an important addition to the toolbox for tackling the problem of optical scattering, and the work brings in new and intriguing thoughts to the research field. I think the work will be of great interest to the readers of Nature Communications. I am happy to recommend the publication of the work.

We thank the reviewer for his positive comments. As an addendum, we would like to stress that an additional merit of our method, in our humble opinion, is its ability to focus and image non-invasively using **linear fluorescence**, that is ubiquitous in imaging but remains also extremely challenging for wavefront-shaping as guidestar mechanism [Horstmeyer, R., Ruan, H., & Yang, C. (2015). Guidestar-assisted wavefront-shaping methods for focusing light into biological tissue. Nature photonics, 9(9), 563].

Here are some comments that the authors may want to consider addressing before publication.

1. On page 3, “In principle, should correspond to the number of independent targets in the system, which is unknown in such reflection configuration.” Does the speckle size on the target play a role here? Does the rank of the matrix depend on the number of the speckles on the fluorescence targets? If so, it may be more accurate to describe the target population in terms of the number of speckles. If there is only one target that covers multiple speckles, what is the rank of the matrix? Can the method achieve a single speckle focusing resolution in this case?

We thank the reviewer for raising this point. Indeed the size of the speckle grain is an important parameter here. In our first experiments with beads, we ensured that the speckle grain size matched the size of one fluorescent bead, i.e. around 1 micron. In this case, each bead acts as a single source, because it is uniformly excited. However this configuration is very specific and we agree with the reviewer that our sentence is therefore not general and lacks accuracy. We thus rephrase the sentence as follows (changes are indicated in **bold**): “*In principle, should correspond to the number of independent **sources** in the system, which **in this specific case corresponds to the number of resolvable targets since the speckle grain matches the size of the bead.** Note that is unknown in such reflection configuration.*”

To be more precise, the rank of the matrix is generally given by the number of separable sources, i.e. the number of speckle grains across the fluorescent object. This is one reason why we extended the study to continuous objects such as the fluorescence-stained pollen seeds (see Figure 5 of the manuscript); the main result being that our technique is robust and still works in this situation. In this case, the object covers multiple speckle grains and the rank of the matrix would be given by the entire fluorescence volume divided by the excitation speckle grain volume.

But, whatever the fluorescent object, focusing should always be diffraction-limited, meaning that we focus the illumination on a single speckle grain, even if there are multiple targets in the region. As an illustration, we display below the speckle focusing on a pollen seed, where the size of the spot is diffraction-limited (the focus appears distorted because the object itself may absorb and diffract light, the image being taken in transmission).

Fig.A: Diffraction-limited focusing on an extended fluorescence-stained pollen seed. (a) Image of the pollen seed in bright field with a white illumination. (b) Focusing using our NMF+PR technique (the image is taken in transmission).

2. It would be helpful if the authors can provide an estimation of the population limit of fluorescence targets under which the image can be successfully reconstructed.

As mentioned in the manuscript, the main limitation of our technique to scale to more complex objects is indeed the contrast of the epi-detected fluorescent speckle. It will limit the maximal number of separable fluorescent sources, or diffraction spots, within the sample. The contrast

decreases with a mild squareroot dependency on the number of sources, meaning for instance multiplying the number of targets by 4 would only decrease the contrast by a factor 2.

However, to increase the overall contrast, it is possible to play with the number of independent spectral bands and polarization. The number of spectral bands is defined as the fluorescence bandwidth divided by the spectral correlation bandwidth of the medium, which is directly related to the scattering medium properties (transport mean free path and thickness). Also, fluorescence is not polarized which reduces the overall contrast by a factor $\sqrt{2}$. We have noticed that the polarization effect was not mentioned in our manuscript. We added the following sentence (changes are indicated in **bold**): “*out (p) corresponds to a low contrast speckle because, first, the fluorescence emission is broadband and **not polarized**, second, the N beads generate N different speckles that partially average out*”.

Furthermore, the ability to separate sources via NMF does not simply depend on the contrast, but also on the SNR of the recorded images and on the number of patterns. In our configuration, we were able to demix at least 30 separate sources with a thin first diffuser (Figure 2) and around 50 diffraction spots for the pollen seeds placed behind a weaker diffuser (Figure 5). This is by no means a maximum: while we indeed tried multiple objects with various complexity, we did not specifically explore this limit in a quantitative way. One of the reasons is that it is not easy to design fluorescent sample with a specific complexity.

For a more quantitative analysis, we may refer to our precedent work on speckle variance optimization [Boniface, A., Blochet, B., Dong, J., & Gigan, S. (2019). Noninvasive light focusing in scattering media using speckle variance optimization. *Optica*, 6(11), 1381-1385.]. In the Supplementary Information we investigated how the estimation of the speckle contrast is affected in presence of noise. We showed that there is a trade-off between the number of sources, the fluorescence bandwidth, the scattering medium properties and the surrounding noise, in order to have sufficient contrast. We believe that this study also applies to our NMF + PR technique, where the speckle contrast limits the fluorescence demixing. We decided to refer to it more explicitly in the discussion part: “*One limitation is the contrast of the measured speckle, that decreases with the complexity (number of separate emitters) of the object, but only with a mild square root dependency. **For further insight, we may refer to the Supplementary Information of [7] where a study about the contrast measurement in presence of noise was carried out and [19] where the effect of noise on the NMF is investigated.***”

We also added a sentence in the discussion about the object complexity: “***Reconstructions up to around 50 focus positions have been performed in this work.***”

As a conclusion, and to answer the referee’s question, we firmly believe that our technique can be scaled to much more complex objects: further refinements and optimization of our technique should allow resolving target numbers well above 100, and to fluorescent objects much more complex than a pollen seed, but this remain of course to be demonstrated experimentally.

3. In the caption of Fig. 1. “A speckle field, $E_{exc} = TE_1$, illuminates the object which emits a fluorescence signal in return.” The dimension of E_1 is not explicitly described. This sentence implies that E_1 is a vector, i.e. one input field. However, in a couple of other places in the manuscript such as in Fig. 1, “ $H = |TE_1|$ ”, E_{in} looks like a matrix consisting of a set of input fields because H is a matrix. It would be great if the authors can clarify the notation.

We thank the reviewer for spotting this discrepancy. In the current manuscript, E_{in} is defined as a matrix that contains all the input masks that will be successively displayed onto the SLM. In the sentence pointed out by the referee, we consider a single realization, meaning that we only take one mask, $E_{in}(p)$, from the complete matrix. To avoid any confusion with the notation, we decided to add the index p all the time we consider a single realization. Therefore, we have rephrased the sentences as follows (changes are indicated in **bold**): “A **given** speckle field, $(\cdot) = (\cdot)$, illuminates the object which emits a fluorescence signal in return, **with** $= , \dots ,$ **the index of the SLM pattern**. A portion of the latter is back-scattered by the medium and epi-detected on a camera, $(\cdot) = |(\cdot)|/2$.”

Note that in the main text the index is not always added since we may also consider the complete matrix like “ $H = |TE_{in}|/2$ ” where $H \in \mathbb{R}^{N \times P}$, $T \in \mathbb{C}^{N \times N_{SLM}}$ and $E_{in} \in \mathbb{C}^{N_{SLM} \times P}$.

4. On page 4, “FIG. 4. Reconstruction examples for continuous and 3D fluorescent objects through a ground glass diffuser.” What is the difference between using 3D and 2D samples? How 3D is defined here? Are the fluorescent objects extended larger than the speckle grain size in the axial direction? If not, the samples may be more precise to be classified as 2D objects in the context of this method.

As all three reviewers raise concerns about 3D imaging, we decided to clarify our statement regarding this point and detail more quantitatively the possibilities of our approach.

As 3D may be too strongly associated with 3D imaging, we modified the manuscript to use the term “volumetric” instead: “Reconstruction examples for continuous **volumetric** fluorescent objects through a ground glass diffuser”. This change emphasizes that fluorescent targets are not restricted to a 2D plane like the fluorescent beads sample, but the technique is still robust enough to focus light and retrieve a 2D projection non-invasively.

Let us discuss whether in Figure 5, the sample can be considered as volumetric. In our manuscript, we define a volumetric sample as a fluorescent object which is larger than the speckle grain size in the axial direction. First, let us demonstrate that the pollen seeds meet this condition. At the object plane, the estimation of the illumination typical speckle grain lateral size gives $1.2 \mu\text{m}$ approximately (see Fig. C). This value has been obtained from the autocorrelation of a speckle pattern recorded in transmission with a collection objective of NA 0.4. We need to emphasize here that this NA is maybe too low to collect the highest spatial frequencies of the illumination speckle, meaning that the speckle grain size may only be smaller than $1.2 \mu\text{m}$. According to Gaussian beam properties, the corresponding Rayleigh range is $z_r = \pi \omega_0^2 n / \lambda = 3.2 \mu\text{m}$ (we took $n = 1.5$ since the pollen seeds are mounted inside microscope slides). The beam extension in both directions is twice this value, which means that in our optical configuration a speckle grain is at most $6\text{-}7 \mu\text{m}$ axially. Although the size of the pollen seeds have not been measured along the z-axis we can suppose that on average they have similar extent in the x, y and z directions. From the ground truth fluorescence images (Figure 5) we can say that they spread over $20 \mu\text{m}$ approximately. Consequently, they are excited by more than one speckle grain along the optical axis and can be considered for this reason as volumetric objects.

To conclude, our approach is naturally 3D for focusing, but we fully agree that the reconstruction based on speckle correlations was only demonstrated in 2D, albeit the object is volumetric. It may be extended to the third dimension, but would require a significant

development to include the axial memory effect (replacing the shift correlation by the addition of a phase curvature) in the model. We will come back on this point later on, when addressing the questions of reviewer 2.

Fig.B: Estimation of the speckle grain lateral size. (a) Speckle image at the plane of the object (taken from the same dataset as Figure 4 of the manuscript). (b) Autocorrelation of the speckle represented in (a). (c) Cross section of the autocorrelation. Its peak is fitted with a Gaussian function whose full width at half maximum is FWHM = 1.69 μm. The average grain size of the speckle pattern shown in (a) is approximately $1.69 / \sqrt{2} = 1.2 \mu\text{m}$.

5. On page 1, “Although imaging fluorescent objects through thin scattering media can be done thanks to the memory effect [17, 18], a general method to focus and image a fluorescent object at depth is still missing.” This sounds like the method here does not require memory effect, but I think the demonstrated method also relies on memory effect for imaging.

We agree with the reviewer that this statement may be misleading and thank him for noticing it. We thus rephrase the sentence as follows (changes are indicated in **bold**): “*Although imaging fluorescent objects through thin scattering media can be done thanks to the memory effect [17, 18], a general method to focus on **fluorescent objects at depth, and image them if they extend beyond the memory effect** is still missing.*”

First, let us remind that focusing on several targets one by one does not require any memory effect here. This is shown in Figure 1 of the manuscript where numerical simulations with matrices entirely random were used to show the principle of the technique (simulation code available here: https://github.com/laboGigan/NMF_PR). As the referee rightly states, our technique for imaging does require some memory effect in order to retrieve the relative positions between points in the image. Previous techniques to image objects inside scattering media, either autocorrelation-based or raster scanning after focusing, have a field-of-view limited by the memory effect range.

The technique we introduce circumvents this limit on the field-of-view. From the information gained during the measurement procedure, we reconstruct the full object exploiting local correlation only (patch by patch), enabling the reconstruction of areas larger than the memory effect range (as demonstrated in Figure 4).

6. It seems the retrieved ingoing and outgoing transmission matrices are independent of each other in the reported method. In terms of hardware implementation, the camera and SLM do not need to be aligned pixel to pixel. However, if the camera and SLM are precisely aligned like the digital optical phase conjugation system, is there any benefit we can gain from this implementation? For example, additional constraints on the transmission matrices for a better reconstruction.

The retrieved ingoing and outgoing matrices are indeed independent. The ingoing transmission matrix (denoted T in the manuscript) is complex valued and can be used for wavefront shaping such as focusing of the illumination at 532 nm. In contrast, the outgoing matrix (denoted W in the manuscript), is at a different wavelength and is an intensity matrix (since fluorescence is a broadband and incoherent process). Thus, we do not expect here any significant correlations between the two matrices that can be exploited in a straightforward way by aligning pixel to pixel. For these reasons, we do not currently see a benefit in implementing pixel-to-pixel alignment. Note that such precise alignment is quite challenging, only mastered by a few teams worldwide. So we would argue that the fact we do not need to align pixel to pixel is definitely a neat feature of our technique, that simplifies it considerably compared to DOPC approaches.

This being said, our method may be generalizable to other contrast mechanisms, for instance Raman, where W and T may have relatively similar wavelengths. In this case, the two transmission matrices would be correlated which could simplify the reconstruction process, and there may be an interest to align pixel to pixel.

7. This method of using NMF to retrieve eigen-patterns shares some similarities with the algorithms used in TROVE (Nature Photonics volume 7, pages300–305 (2013)) and AOTM (ref 12), which are also used to retrieve “eigen-patterns”. Could the authors elaborate on the common and different points of these algorithms?

We thank the reviewer for pointing this connection between our approach and TROVE/AOTM techniques. While we are familiar with this literature, we did not indeed consider the connection with the present work. In these works, the sample is illuminated with a set of input wavefronts and ultrasound tagged speckle fields that exit the sample are recorded and stored in a matrix. While different in nature, TROVE and AOTM have in common the computation of the singular value decomposition (SVD) of a matrix (or its covariance in TROVE, which is related) in order to achieve optical resolution. The SVD gives access to a set of eigenvectors. The first singular vector allows speckle-size focusing at a single point located at the center of the ultrasonic focus.

Our NMF algorithms also tries to find the singular vectors and values of a matrix, but the matrix does not correspond to the same quantities, and the algorithm operates under different priors not present in TROVE/AOTM, namely positivity, which tends to provide sparse solutions and higher resilience to noise. The positivity constraint makes the NMF well adapted to incoherent processes while TROVE and AOTM requires coherence. This means in particular that no holographic detection is required since we directly use the intensity fluorescent speckles. Instead, an additional step of phase retrieval is needed to recover the complex-valued transmission matrix and focus light on the sample. Lastly, all the eigen-patterns from the NMF (and not only the first one as in TROVE) can be used to focus light to diffraction-limited spots.

This being said, TROVE and the AOTM have a very different scope and other advantages, since they rely on acoustic tagging and can work subsequently with any optical contrast mechanism.

To sum up our answer to the reviewer, we agree that our technique presents similarities with TROVE and AOTM strategies. In our current article, we only cited the AOTM article. We agree it is important to also cite the TROVE paper, and added a sentence (in **bold** at the end of the following paragraph): “*This corresponds exactly to the framework of Non-negative Matrix Factorization (NMF) that we therefore use to estimate W and H from I_{out} , see Fig. 1b. Thanks to its robustness and interpretability, this framework has already been applied in many settings [19], including the fluorescent readout of neuronal activity [20, 21], but has never been associated with wavefront shaping yet. **It is interesting to note that similar approaches based on leading eigenvector decompositions (without the non-negativity assumption) have already been used in optics [Dong, J., Krzakala, F., & Gigan, S. (2019, May). Spectral method for multiplexed phase retrieval and application in optical imaging in complex media. In ICASSP 2019-2019 IEEE (pp. 4963-4967)] and in combination with acoustics [Judkewitz, B., Wang, Y. M., Horstmeyer, R., Mathy, A., & Yang, C. (2013). Speckle-scale focusing in the diffusive regime with time reversal of variance-encoded light (TROVE). Nature photonics, 7(4), 300-305., 12] for imaging at depth.**”*

Author’s Response To Reviewer 2

The authors describe a method to image fluorescent objects hidden behind a scattering medium by using only epi detection. This is a topic of high importance for optical imaging in tissue. A previous attempt by Mosk and colleagues for imaging using only epi detection was limited to the memory effect of the medium, which can be very small.

Here the authors propose a method which goes beyond the memory effect range. This is quite an advance in the field.

It is in many ways quite different from the said previous attempt. The detection uses a 2D camera instead of a single pixel detector and more importantly makes use of the fluorescent speckle pattern.

The method is novel and merit publication.

The manuscript is very well written and thorough.

1. The fluorescent beads are all placed in the same plane. What happened if they were placed at different depth? This is a situation that would occur in practice for imaging an unknown sample. The authors hint that the method could be used in 3D.

In our first experimental demonstrations (Figure 2-3-4 of the manuscript) the object constituted of several beads is 2D: all the beads lie in a plane orthogonal to the optical axis. But in theory, the fluorescence demixing with the NMF should also work if the beads are distributed in 3D as long as the beads emit different eigen-patterns. However we see two situations that may be problematic.

1-A fluorescent object made of two beads, one on top of the other (i.e. in two different z planes but along the same axis). Within the memory effect axial range, the corresponding eigen-patterns should be very similar up to a magnification factor, and it remains to be verified if the difference is large enough for the NMF to separate the two contributions.

2-A 3D fluorescent object extending over a wide depth range. In this case the sources close to the surface will be more excited and the backscattered fluorescence much more intense than sources located deeper inside the medium. The contribution of each eigen-pattern to the overall speckle will therefore depend on the depth of the corresponding source. We may suppose that at some point if the emitter is too far from the surface, its participation to the overall epi-detected fluorescent speckle will be very weak and our algorithm may not be able to either focus on or image it.

In our last experiment with pollen seeds that are 3D objects for our system (Figure 5 of the manuscript) we show that we are able to retrieve some information on their shape. However the reconstruction is only 2D, since the algorithm is based on a 2-dimensional model of the object and only recovers lateral shifts. For a volumetric object, it still recovers a 2D projection of the 3D object. One possibility would be to develop more complex algorithms that are able to track the magnification of the speckle and gives access to the position of the source in 3D as in [Antipa, Nick, et al. "DiffuserCam: lensless single-exposure 3D imaging." *Optica* 5.1 (2018): 1-9.] with a surface diffuser. We can also think of changing the acquisition process and for instance moving the camera to retrieve additional information about z as in [Okamoto, Y., Horisaki, R., & Tanida, J. (2019). *Noninvasive three-dimensional imaging through scattering media by three-dimensional speckle correlation. Optics letters*, 44(10), 2526-2529.].

The following paragraph has been added in the main text to make this point more precise: ***“It is important to note we only reconstruct a 2D projection of an object, even though the object might be volumetric. Strategies for 3D reconstruction in scattering environments have been proposed [32,33], but require specific experimental configurations. Additionally, techniques to reduce background fluorescence [30,31] may be useful to reduce the necessity of a full 3D reconstruction.”***

2. Wouldn't such a linear technique be limited to the surface of the fluorescing tissue because of the double pass (loss of incident intensity reaching deeper layers and loss of fluorescent intensity on the way back)?

We thank the reviewer for highlighting this possible issue. We agree that loss of intensity may decrease the SNR and limit the penetration depth. As a first remark, in the multiple scattering regime, the transmitted intensity only decreases linearly with penetration depth (unlike ballistic light which decreases exponentially), therefore quite a significant portion still reaches a deep fluorescent object and similarly on the way back to the camera.

To develop on this point, if the fluorescence object is uniformly tagged in volume, the outer layer will contribute more to the speckle and this will prevent reconstruction from the deeper layers. However, this argument may also be raised for a variety of other techniques in fluorescence microscopy. Since our method also requires a relative sparsity of the object, a way to circumvent this issue in practical applications would to specifically tag objects in a small region of interest, only where we want to image.

3. It seems that the power of the technique is that it focuses all the incident light on a single bead. Would confocal detection using ballistic return photons be possible?

With volumetric scattering, the number of ballistic photons exponentially decreases with depth. Their number becomes negligible after a few transport mean free paths. Thus, confocal detection would be impractical to image at depth.

We must emphasize here that in our system, there should be no ballistic photon. The diffusers are designed to prevent the ballistic component of both the incoming excitation and the retro-reflected fluorescence. Still, if some ballistic light remains, we expect our NMF + PR pipeline to work as it can be applied with any linear optical system. But, in order to use it in conjunction with a confocal detection, one would need to know to conjugate the position of the pinhole with the location of the focus, which is unknown in our pipeline.

Author's Response To Reviewer 3

The manuscript by Boniface and coworkers follows in the steps of a previous manuscript by the same group (Ref. 7) which showed the utility of characterization and maximization of the contrast of the low-coherence fluorescence speckle as a means to focus light in scattering media. Now the authors replace the adaptive search for focusing on individual emitters by a non-negative matrix factorization approach which enables retrieval of the transmission matrix of both the ingoing coherent light and the outgoing fluorescence (at least for the center wavelength for the latter). The beauty of this approach is that it enables to perform the measurements in a wide field, reducing dramatically the time, and cleverly utilizing the multiple fluorescent emitters as witnesses in the measurement of the excitation field transmission matrix. As this approach is new and has exciting potential consequences, it can warrant publication in Nature Communications provided that the authors address the following points:

1. The explanation in the introduction is written in a confusing way. The measured matrix H is low rank because the fluorescent "witnesses" are sparse. reconstruction of the transmission matrix thus seems like magic. It is only much later in the text that the authors actually cite the number of SLM patterns required to perform the retrieval of the transmission matrix. While this appears in Fig. 1c, it should be much more explicitly stated already when describing the method since it is probably the most significant issue.

We would like to thank the reviewer for noting that indeed, a detailed discussion on the phase retrieval algorithm was only found in the end of the paper and the supplementary information. We have added a more detailed explanation of the algorithm earlier, at the end of page 2 (modifications in **bold**):

*"In a second step, a phase retrieval (PR) algorithm allows retrieving the ingoing field- T_M , T [22, 23], from H . **This computational problem has been studied extensively for the case of***

random matrices and is in theory solvable when the number of measurements (here, P) is a few times larger than the dimension of the unknown (here, N) [26-28].”

2. Relating to the same point, could the authors compare the retrieved images as a function of the number of degrees of freedom in the SLM used in getting the transmission matrix? This would correspond to the tradeoff between measurement time and output quality which is an important parameter.

We thank the reviewer for this important question. We did study the effect of the ratio of the measurement numbers to the number of pixels (parameter α) in Fig 3 in the Supplementary Information, but did not explicitly investigate the dependency on the number of pixels. Indeed, the number of measurements also scales linearly with the number of controlled SLM pixels. It is thus important to have a good estimate of the SLM dimension for imaging applications.

While the first experiments are done with $N_{\text{SLM}} = 1024$ pixels, the reconstruction of the pollen seeds in Figure 5 only makes use of $N_{\text{SLM}} = 256$ pixels. This number decreases our focusing ability, since we know that the SNR of the focused spot is proportional to N_{SLM} , but also speeds up the acquisition (4-times less patterns are required). With $N_{\text{SLM}} = 256$ pixels, the SNR of the focus is not very high, around 10-20 (see Fig.A of this document) but sufficient to connect the epifluorescence detected fluorescent speckle to the eigen-pattern of the focused emitter. Therefore, an image of the object by cross-correlating these measured speckles is possible (see Fig.C a-b-c).

Fig.C: Pollen seeds reconstruction for a different number of controlled SLM modes. (a)-(b)-(c) reconstruction with $N_{\text{slm}}=256$, (d) reconstruction with $N_{\text{slm}}=1024$. (e)-(f)-(g)-(h) Ground truth fluorescence images.

To show that the image quality is not drastically affected, we conducted an additional experiment on another pollen seed with more input modes, $N_{\text{SLM}} = 1024$ (see Fig.C d). As one can notice the four reconstructed images have more or less the same quality and we can conclude that in this situation 4 times more measurements is not so advantageous. Hence, one may start with a relatively low number of SLM pixels, of the order of a few hundreds, to speed up the Transmission Matrix measurement.

Although we show qualitatively that the shape of the pollen seeds can be retrieved, their reconstructions are imperfect when comparing with the ground truth images that are taken with

a different objective. Therefore it is relatively difficult to define a precise metric to quantify the quality of the reconstruction and infer on the impact of .

To be more precise in the manuscript, we added the following sentence to the discussion: *“With = 56 pixels, the SNR of the focus is not very high, around 10-20, but sufficient to connect the epi-detected fluorescent speckle to the eigen-pattern of the focused emitter and reconstruct the shape of the pollen seeds. Increasing the number of SLM pixels to = 104 did not improve significantly the results.”* and replaced Figure 5 of the manuscript by Fig. C, reported above and modified accordingly its caption.

3. The authors discuss in detail the dependence on the emitter number (and the square root decline in speckle contrast). They do not discuss the effect of the bandwidth of emitted fluorescence. If anyone is to use this type of method, it should be clearly stated (preferably quantitatively) how much the performance deteriorates for dyes or stains with broader fluorescent emission.

As already mentioned in our answer to question 2 of reviewer #1, the contrast of the overall speckle indeed depends also on the number of spectral bands (defined as the bandwidth of the fluorescence divided by the spectral correlation of the speckle). Just like for the number of sources, the contrast will only have a squareroot dependency on this number. In our system the fluorescence emitted by the beads has a bandwidth of ~50 nm (see Fig.D), which in our case of thin scattering medium does not correspond to more than one spectral band.

Fig.D: Wavelengths at stake in the system. The laser, a narrow band peaked around 532 nm, excites the fluorescent beads (with broad absorption and emission spectra). Part of the one-photon fluorescence is epi-detected after passing through a dichroic mirror at 550 nm and a longpass filter at 532 nm. This plot was done with SearchLight.

For a broader emission spectrum (or for a thicker medium with a narrower spectral correlation), the contrast decreases with a squareroot scaling: if the bandwidth is twice broader, the contrast is only decreased by a factor $\sqrt{2}$.

On the one hand, if the contrast is the limiting factor, we would like to mention that one may introduce a narrow passband spectral filter or a polarizer to increase the overall contrast. We

decided to highlight this possibility when we mention the elements of the optical setup in the section Methods of our manuscript (modifications in **bold**): “*We use a dichroic mirror shortpass 550 nm (Thorlabs) and two other filters (F): a 532 nm longpass (Semrock) and a 533 nm notch (Thorlabs). **Additional spectral filters can be used before light reaches CAM1 in order to narrow the spectral width of the detected fluorescence and increase the contrast of the fluorescent speckle. To the same end, a polarizer can be inserted, enhancing the contrast by a factor $\sqrt{2}$.***”

On the other hand, if the amount of signal is the limiting factor (due to bleaching or for fast acquisition), filtering would reduce the photon budget. In short, it depends on the situation but definitely spectral filtering can increase the overall contrast.

4. As can be seen from Fig. 4 (and since the authors assume low rank matrices) the method inherently introduces graininess in the retrieved images even when it is not actually present in the samples. The authors should at least comment on this. Better is if they have a solution to differentiate between an intrinsically grainy sample (like the beads) and a smooth one.

We agree that the images look grainy and thank the reviewer for observing this. It leads to an interesting discussion: fundamentally, any Transmission Matrix approach is discrete, and our low-rank approach emphasizes this point. From a relatively limited number of focal points, we are able to localize their relative positions exploiting local memory effect correlations. We believe this cross-correlation step is necessary to retrieve an image with a field-of-view larger than the memory effect range.

Still, it should be possible to perform raster scanning around each focus position, following the principle of techniques using the total linear fluorescence only [Vellekoop, I. M., & Aegerter, C. M. (2010). *Scattered light fluorescence microscopy: imaging through turbid layers. Optics letters, 35(8), 1245-1247.*]. The field-of-view would be limited to the memory-effect range around each focus position, but the retrieved image would be smooth and continuous. This information could then be combined with the cross-correlation image to improve the imaging quality and field-of-view with continuous objects.

We decided to add the following sentence in the text to mention this point: “***While the reconstruction appears grainy, we do retrieve the main and brightest features of the pollen seeds.***”

5. All the objects used here are thin objects behind non-fluorescent diffusers. Thus, they do not represent a full 3D problem. Are the claims that this can really work in 3D samples really substantiated?

The problem of 3D objects has already been discussed in question 4 of reviewer #1 and question 1-2 of reviewer #2. We invite the reviewer to refer to them for a more detailed explanation.

We agree that we do not retrieve a 3D image, but show that the technique is robust enough to work with volumetric objects (to focus excitation light and retrieve 2D projections of the objects). The manuscript has been modified accordingly to clarify this important point.

REVIEWERS' COMMENTS:

Reviewer #1 (Remarks to the Author):

The authors adequately addressed reviewers' comments and further improved the already well-written manuscript. The claim on the ability to focus on and image linear fluorescence is sound and valid. I am happy to recommend publication of the manuscript in the current technical form.

Reviewer #2 (Remarks to the Author):

The authors have responded my questions thoroughly and have clarified the points raised. I am happy to recommend publication.

Reviewer #3 (Remarks to the Author):

The revision by boniface and coworkers addresses the referee concerns adequately. The manuscript can be published in Nature Communications in its present form.